# Accelerating Nash Equilibrium Convergence in Monte Carlo Settings Through Counterfactual Value Based Fictitious Play

**Ju Qi**[1,2,†]**, Hei Falin**[1,2]**, Feng Ting**[1,2]**, Yi Dengbing**[1,2]**, Fang Zhemei**[1,2,†]**, Luo Yunfeng**[1,2]

1. School of Artificial Intelligence and Automation, Huazhong University of Science and Technology
2. National Key Laboratory of Science and Technology on Multispectral Information Processing
`{juqi, heifalin, fenting, yidengbing, zmfang2018, yfluo}@hust.edu.cn`
[†] Corresponding author.

## Abstract

Counterfactual Regret Minimization (CFR) and its variants are widely recognized as effective algorithms for solving extensive-form imperfect information games. Recently, many improvements have been focused on enhancing the convergence speed of the CFR algorithm. However, most of these variants are not applicable under Monte Carlo (MC) conditions, making them unsuitable for training in large-scale games. We introduce a new MC-based algorithm for solving extensive-form imperfect information games, called MCCFVFP (Monte Carlo Counterfactual Value-Based Fictitious Play). MCCFVFP combines CFR's counterfactual value calculations with fictitious play's best response strategy, leveraging the strengths of fictitious play to gain significant advantages in games with a high proportion of dominated strategies. Experimental results show that MCCFVFP achieved convergence speeds approximately 20%~50% faster than the most advanced MCCFR variants in games like poker and other test games.

## 1 Introduction

Game theory investigates mathematical models of strategic interactions among rational agents, aiming to identify a Nash Equilibrium (NE) where no participant can gain by deviating from the established strategy. For complete information games, it is possible to segment the game into smaller sub-games and apply backward induction to determine the equilibrium. In contrast, in incomplete information games, the inability to directly apply sub-game payoffs in the backward induction algorithm significantly increases the complexity of finding an equilibrium.

The Counterfactual Regret Minimization (CFR) algorithm Zinkevich *et al.* [2007] is crucial for solving extensive-form games with incomplete information. CFR has numerous variants, such as CFR+ Tammelin [2014], Lazy CFR Zhou *et al.* [2018], PCFR Farina *et al.* [2021], DCFR Brown and Sandholm [2019a], and PDCFR Xu *et al.* [2024]. Nevertheless, these variants typically require a full traversal of the entire game tree, which is impractical for real-world applications. In contrast, games like no-limit Texas hold'em, DouDizhu, and Mahjong have $10^{162}$ Moravík *et al.* [2017], $10^{83}$ Zha *et al.* [2021], and $10^{121}$ Li *et al.* [2020] information sets, respectively, making full traversal infeasible.

To address this, Lanctot et al. Lanctot *et al.* [2010] proposed Monte Carlo CFR (MCCFR), which reduces the number of game tree nodes visited in each iteration through sampling. This makes MCCFR the preferred algorithm for training large-scale games. However, most CFR variants cannot improve the convergence speed in the case of MCCFR. Only a few variants, such as Discount MCCFR Brown and Sandholm [2019a] and VR-MCCFR Schmid *et al.* [2019], are currently suitable

for MCCFR. Therefore, it is crucial to study the characteristics of large-scale games and how to leverage these characteristics to accelerate the convergence of MC-based game learning algorithms.

We believe that the Fictitious Play (FP) algorithm is highly suitable for addressing large-scale incomplete information games, due to the characteristics of using the best response (BR) strategy. FP was first introduced in Brown's 1951 article Berger [2007]; Brown [1951]. The treatise *The Theory of Learning in Games* Fudenberg and Levine [1998] consolidated prior research, setting a standardized framework for FP . Generalized Weakened Fictitious Play (GWFP) Leslie and Collins [2006] further established that under specific perturbations and errors, convergence to NE is attainable in a manner consistent with FP . Hendon et al. Hendon *et al.* [1996] expanded FP to the domain of extensive-form games. The development of the Full-Width Extensive-Form Fictitious Play (XFP) algorithm Heinrich *et al.* [2015], predicated on GWFP, has facilitated faster convergence in these games.

Our algorithm, Monte Carlo Counterfactual Value-Based Fictitious Play (MCCFVFP), optimizes FP in extensive-form games by incorporating counterfactual value into the BR strategy calculations. We have theoretically proven that MCCFVFP can converge to a NE, and experimentally demonstrated that in large-scale games, MCCFVFP fully leverages the high proportion of dominated strategies to achieve faster convergence speeds than MCCFR. Our code can be found at GitHub.

## 2 Notation and Preliminaries

### 2.1 Game Theory

#### 2.1.1 Normal-Form Game

The normal-form game serves as the foundational model in Game Theory. Let $\mathcal{N} = 1, 2, \ldots, i, \ldots$ represent the set of players, where each player $i$ has a finite action set $\mathcal{A}^i$. Player $i$'s strategy, denoted $\sigma^i$, is a probability distribution over $\mathcal{A}^i$ and is represented by a $(|\mathcal{A}^i| - 1)$-dimensional simplex. Here, $|\cdot|$ indicates the cardinality of the set, and $\sigma^i(a')$ denotes the probability with which player $i$ selects action $a'$. Let $\Sigma^i$ denote the strategy set for player $i$, such that $\sigma^i \in \Sigma^i$. A strategy profile, $\sigma = \prod_{i \in \mathcal{N}} \sigma^i$, represents the collection of strategies for all players, while $\sigma^{-i} = (\sigma^1, \ldots, \sigma^{i-1}, \sigma^{i+1}, \ldots)$ includes all strategies in $\sigma$ except for that of player $i$. The entire set of strategy profiles is denoted by $\Sigma = \prod_{i \in \mathcal{N}} \Sigma^i$, where $\sigma \in \Sigma$. The payoff function for player $i$, defined as $u^i : \Sigma \to \mathbb{R}$, is finite. The notation $u^i(\sigma^i, \sigma^{-i})$ indicates the expected payoff to player $i$ when they select the pure strategy $\sigma^i$ and all other players follow the strategy profile $\sigma^{-i}$. Finally, define the payoff interval of the game as $L = \max_{\sigma \in \Sigma, i \in \mathcal{N}} u^i(\sigma) - \min_{\sigma \in \Sigma, i \in \mathcal{N}} u^i(\sigma)$.

#### 2.1.2 Extensive-Form Games

In extensive-form games, typically represented as game trees, the player set is $\mathcal{N} = 1, 2, \ldots$. Each node $s$ represents a possible state in the game, forming a set $\mathcal{S}$, with terminal states represented by leaf nodes $z \in \mathcal{Z}$. At each state $s \in \mathcal{S}$, the action set $\mathcal{A}(s)$ includes all possible actions available to a player or chance. The player function $P : \mathcal{S} \to \mathcal{N} \cup c$ assigns each state an acting party, with $c$ indicating a chance event. Information sets $I \in \mathcal{I}^i$ group states that player $i$ cannot distinguish among, reflecting their uncertainty. The payoff function $R : \mathcal{Z} \to \mathbb{R}^{|\mathcal{N}|}$ maps each terminal state to a payoff vector for the players. For each information set $I \in \mathcal{I}^i$, the behavioral strategy $\sigma^i(I) \in \mathbb{R}^{|\mathcal{A}(I)|}$ defines a probability distribution over available actions.

#### 2.1.3 Nash Equilibrium

The BR strategy of player $i$ to their opponents' strategies $\sigma^{-i}$ is

$$b^i(\sigma^{-i}) = \arg\max_{a^i \in \mathcal{A}^i} u^i(a^i, \sigma^{-i}), \tag{1}$$

although a mixed strategy may also become a BR strategy, it is much easier to find a pure BR strategy than a mixed BR strategy in engineering implementation, so the BR strategies discussed in this article are all pure BR strategies. If there are multiple BR strategies at the same time, one of them will be returned randomly. Define the exploitability $\epsilon^i$ of player $i$ in strategy profile $\sigma$ as:

$$\epsilon^i = u^i(b^i(\sigma^{-i}), \sigma^{-i}) - u^i(\sigma), \tag{2}$$

and total exploitability as $\epsilon = \sum_{i \in \mathcal{N}} \epsilon^i$. A Nash equilibrium is a strategy profile $\sigma$ such that $\epsilon = 0$.

During iterations, if the strategy's exploitability satisfies $\epsilon \propto T^{-\frac{1}{2}}$, the convergence rate of the algorithm is $\mathcal{O}\left(T^{-\frac{1}{2}}\right)$. Additionally, the time complexity within one iteration is a critical factor affecting the convergence rate. We use $\mathscr{O}(\cdot)$ to describe the time complexity needed for one iteration.

### 2.1.4 Dominated Strategy and Clear Games

In game theory, strategic dominance occurs when one action (or strategy) is better than another action (or strategy) for one player, regardless of the opponents' actions. A classic example is the *Prisoner's Dilemma*, where choosing to **testify** always yields a higher payoff than **staying silent**, regardless of the opponent's decision. Here, staying silent is a dominated strategy. Formally, For player $i$, if there is a pure strategy $a^{i*}$ and a strategy $\sigma^{i*} \in \Sigma^i$ satisfies:

$$u^i(a^{i*}, \sigma^{-i}) \leq u^i(\sigma^{i*}, \sigma^{-i}), \forall \sigma^{-i} \in \Sigma^{-i}, \tag{3}$$

then the pure strategy $a^{i*}$ is a dominated strategy of player $i$. Rational players will invariably avoid choosing a dominated strategy, allowing the elimination of dominated strategy from the action set $\mathcal{A}^i$ without impacting the game's NE Samuelson [1992].

In our analysis, the proportion of dominated strategies in a game is crucial, significantly impacting the convergence speed of various algorithms. We categorize games into two distinct types based on this factor. Games where the number of dominated strategies is less than $\sqrt{|\mathcal{A}|}$ are classified as **clear games**. This term captures a strategic landscape where the preferable choices are evident, owing to the relatively few dominated strategies. On the other hand, **tangled games** are defined by having a number of dominated strategies greater than $\sqrt{|\mathcal{A}|}$, reflecting a more complex and nuanced strategic environment with less obvious choices. The reason for defining these categories in this way can be found in Appendix B.3. Such classification aids in comprehending a game's nature and assessing the effectiveness of different algorithmic approaches.

### 2.2 Regret Matching and Counterfactual Regret Minimization

In normal-form games, let $\sigma_t^i$ be the strategy used by player $i$ on round $t$. The regret of player $i$'s action $a^i$ at time $T$ is:

$$R_T^i(a^i) = \sum_{t=1}^{T} u^i\left(a^i, \sigma_t^{-i}\right) - u^i\left(\sigma_t\right). \tag{4}$$

The new strategy is produced by:

$$\sigma_{T+1}^i(a^i) = \begin{cases} \frac{R_T^{i,+}(a)}{\sum_{a \in \mathcal{A}^i} R_T^{i,+}(a)} & \text{if } \sum_{a \in \mathcal{A}^i} R_T^{i,+}(a) > 0 \\ \frac{1}{|\mathcal{A}^i|} & \text{otherwise,} \end{cases} \tag{5}$$

where $R_T^{i,+}(a) = \max\left(R_T^i(a), 0\right)$. Since the probability of selecting action $\sigma_{t+1}^i(a)$ is proportional to the non-negative regret value $R_T^{i,+}(a)$ for this action, this algorithm is referred to as the regret matching algorithm. Define the average strategy as $\bar{\sigma}_T = \frac{1}{T} \sum_{t=1}^{T} \sigma_t$, which converges to the NE as $T \to \infty$, with a convergence rate of $\mathcal{O}\left(L\sqrt{|\mathcal{A}|/T}\right)$. In each iteration, RM requires a matrix multiplication operation, resulting in an iteration time complexity of $\mathscr{O}\left(|\mathcal{A}|^2\right)$.

CFR is a variant of RM specifically adapted for extensive-form games. The counterfactual value $u(I, \sigma)$ is defined as the expected payoff assuming that information set $I$ is reached and all players follow strategy $\sigma$, with the exception that player $i$ acts specifically to reach $I$. For each action $a \in \mathcal{A}^i(I)$, let $\sigma|_{I \to a}$ represent a strategy profile identical to $\sigma$ except that player $i$ consistently chooses action $a$ within information set $I$. The counterfactual regret is then given by:

$$R_T^i(I, a) = \sum_{t=1}^{T} \pi_{\sigma_t}^{-i}(I)\left(u^i\left(I, \sigma_t|_{I \to a}\right) - u^i\left(I, \sigma_t\right)\right), \tag{6}$$

where $\pi_{\sigma_t}^{-i}(I)$ is the probability of information set $I$ occurring if all players (including chance, except $i$) choose actions according to $\sigma_t$. Let $R_T^{i,+}(I, a) = \max\left(R_T^i(I, a), 0\right)$, the strategy at time $T + 1$ is:

$$\sigma_{T+1}^i(I, a) = \begin{cases} \frac{R_T^{i,+}(I,a)}{\sum_{a \in \mathcal{A}(I)} R_T^{i,+}(I,a)} & \text{if } \sum_{a \in \mathcal{A}^i} R_T^{i,+}(I, a) > 0 \\ \frac{1}{|\mathcal{A}(I)|} & \text{otherwise.} \end{cases} \tag{7}$$

The average strategy $\bar{\sigma}_T^i$ for an information set $I$ on iteration $T$ is:

$$\bar{\sigma}_T^i(I) = \frac{\sum_{t=1}^T \pi_{\sigma_t}^i(I)\sigma_t^i(I)}{\sum_{t=1}^T \pi_{\sigma_t}^i(I)}. \tag{8}$$

Eventually, $\bar{\sigma}_T$ will converge to NE with $T \to \infty$, the convergence rate of CFR is $\mathcal{O}\left(L|\mathcal{I}|\sqrt{|\mathcal{A}|/T}\right)$ and the time complexity of one iteration of the CFR algorithm is $\mathscr{O}\left(|\mathcal{A}||\mathcal{I}|\right)$.

There are many forms of sampling MCCFR, but the most common is external sampling MCCFR (ES-MCCFR) Brown [2020] because of its simplicity and powerful performance. In ES-MCCFR, some players are designated as traversers and others are samplers during an iteration. Traversers follow the CFR algorithm to update the regret and the average strategy of the experienced information set. On the rest of the sampler's nodes and the chance node, only one action is explored (sampled according to the player's strategy for that iteration on the information set), and the regret and the average strategy are not updated.

## 2.3 Fictitious Play

FP assuming all players start with random strategy profile $\bar{\sigma}_{t=1}$, the strategy profile is updated following the iterative function:

$$\bar{\sigma}_{t+1} = (1 - \alpha_t)\bar{\sigma}_t + \alpha_t\sigma_{t+1}, \tag{9}$$

where in vanilla FP $\alpha_t = 1/(t+1)$, $\sigma_{t+1} = b(\bar{\sigma}_t)$. The convergence rate of vanilla FP in a two-player zero-sum game may be $\mathcal{O}\left(\sqrt{T}\right)$ Daskalakis and Pan [2014], and the time complexity of one iteration of the vanilla FP algorithm is $\mathscr{O}\left(|\mathcal{A}|^2\right)$. To speed up iteration, define the average Q-value of player $i$'s action $a^i$ at time $T$ as:

$$\bar{Q}_T^i(a^i) = \frac{1}{T}\sum_{t=1}^T u^i\left(a^i, \sigma_t^{-i}\right). \tag{10}$$

If the game is a two-player game, or if $u^i$ is an affine function (e.g., in a potential game), we have:

$$\begin{aligned}
\bar{Q}_T^i\left(a^i\right) &= \frac{1}{T}\sum_{t=1}^T u^i\left(a^i, \sigma_t^{-i}\right) = u\left(a^i, \frac{1}{T}\sum_{t=1}^T \sigma_t^{-i}\right) \\
&= u^i\left(a^i, \bar{\sigma}_t^{-i}\right),
\end{aligned} \tag{11}$$

therefore, we can conclude that

$$b^i(\sigma^{-i}) = \max_{a^i \in \mathcal{A}^i} \bar{Q}_T^i\left(a^i\right). \tag{12}$$

In 2.1.3, it is noted that BR strategy $\sigma_t^i$ in FP is a pure strategy. As a result, calculating $u^i(a^i, \sigma_t^{-i})$ involves simply selecting a row (or column) from the payoff matrix, eliminating the need for matrix multiplication as required in RM. Consequently, the time complexity of each iteration in the FP algorithm decreases to $\mathscr{O}(|\mathcal{A}|)$, representing a significant improvement over the $\mathscr{O}(|\mathcal{A}|^2)$ complexity in RM.

## 3 Motivation of CFVFP

Currently, the mainstream approach for large-scale games is a combination of "pre-trained blueprint strategy + real-time search." For example, Pluribus initially employs the MCCFR algorithm to establish a blueprint strategy, which it applies in the early stages of the game, and then transitions to real-time search as the game progresses Brown and Sandholm [2019b]. Similarly, in the game of Go, AlphaGo primarily used reinforcement learning to train its policy and value networks, incorporating a limited number of MCTS rollouts during training. However, in actual matches against human experts, AlphaGo significantly increased the number of MCTS rollouts to enhance its real-time decision-making Silver *et al.* [2016, 2017]. This phenomenon can be understood from two perspectives:

1. The complexity of real-world games is so high that it is impossible for any single strategy (even with deep networks) to perfectly handle all game scenarios. Due to this extreme complexity, only sampling-based training methods are feasible. Many current improvements to CFR, such as CFR+ and Predictive CFR, are designed for full traversal of the game tree. While these methods accelerate convergence, they are unsuitable for sampling-based approaches because the inexact payoffs introduced by sampling significantly disrupt the convergence direction of each strategy update.

2. Training a blueprint strategy usually starts with sampling from the initial nodes, resulting in a higher probability of exploring early game nodes compared to leaf nodes. Consequently, while the blueprint strategy might be less effective in the later stages of the game, it often rapidly converges towards the optimal solution in the early stages due to thorough exploration.

After clarifying the specific ideas for training large-scale game AI. Next, we need to think about the characteristics of large-scale games and how to design corresponding MC-type solving algorithms based on these characteristics.

One prominent characteristic of large-scale games is that the majority of strategies are dominated. For instance, in chess and Go, top AI systems and human experts often opt for fixed openings. Similarly, professional poker players tend to fold most hands in the early rounds. Theoretically, DeepMind likens game strategies to a spinning top Czarnecki *et al.* [2020], implying that only a limited subset of strategies can be considered non-dominated. MCCFR experiments show that up to 96% of strategies can be pruned in certain games Lanctot *et al.* [2010]. Likewise, the supplementary materials of Pluribus Brown and Sandholm [2019c] reveal that around 50% of information sets are never encountered during training (indicating that unplayed strategies are almost certainly dominated, with many traversed ones also being dominated). In other words, while it cannot be proven with absolute certainty, the choices of professional players and the experimental outcomes from current advanced AIs suggest that the larger the game, the higher the likelihood that it can be classified as a "clear game."

Additionally, our toy experiment clearly demonstrates that the FP algorithm is more effective for clear games compared to the RM algorithm 1. We attribute this advantage to the fact that FP employs the BR strategy rather than the RM strategy during iterations, which is why we aim to incorporate this feature into the iterations of CFR.

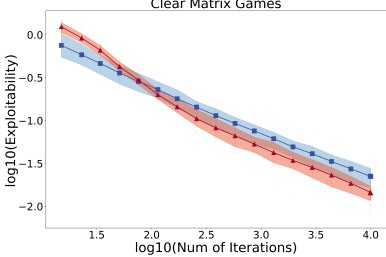
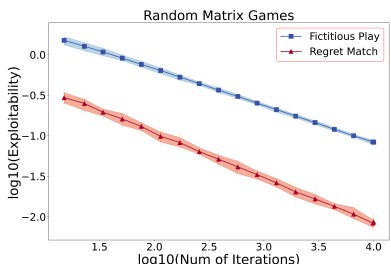

Figure 1: The figure compares the convergence rates of the RM and FP algorithms in a $100 \times 100$ random payoff matrix game generated from a $N(0, 1)$ distribution. In right figure, the convergence for a standard random payoff matrix is shown, while left figure illustrates the convergence in $100 \times 100$ random payoff matrix where the payoffs for actions 1 to 10 are uniformly increased by 5 (causing actions 11 to 100 to have a high probability of being dominated strategies). It can be observed that in this setting, the convergence rate of the FP is very close to that of RM. Considering that the complexity of one FP iteration is only $\mathscr{O}(|\mathcal{A}|)$ compared to the complexity of RM, which is $\mathscr{O}(|\mathcal{A}|^2)$, in a clear game, the overall convergence rate of FP can actually surpass that of RM. Each scenario tested an average of 30 rounds. The shaded areas represent the 90% confidence intervals for these trials. The experiments in Appendix A can also confirm our view from another perspective.

## 4 CFVFP Method

### 4.1 Counterfactual Value Fictitious Play Implementation

Our method can be easily adapted from CFR. First, there is no need to compute the counterfactual regret; instead, we define the counterfactual value:

$$Q_t^i(I, a) = Q_{t-1}^i(I, a) + \pi_{\sigma_t}^{-i}(I) u^i\left(I, \sigma_t|_{I \to a}\right). \tag{13}$$

Second, the strategy in the next iteration is a BR strategy rather than RM strategy:

$$\sigma_{t+1}^i = \arg\max_{a \in \mathcal{A}^i(I)} Q_t^i(I, a). \tag{14}$$

The main distinction between $Q_t^i$ and $R_t^i$ is that $Q_t^i$ omits the average payoff term $u^i(I, \sigma_t)$, as this term does not affect identifying the maximum value of $Q_t^i$. This omission leads to a significant reduction in computation time. Since $\arg\max$ selects the action with the highest counterfactual value, the resulting strategy in each iteration is a pure strategy, similar to the FP algorithm. Thus, we refer to this algorithm as Counterfactual Value-Based Fictitious Play. Additionally, the time complexity of calculating the BR strategy is notably lower than that required for computing the RM strategy.

Finally, since the update $\sigma_{t+1}^i$ is a pure strategy, $\pi_{\sigma_t}^{-i}(I)$ is either 0 or 1. This significantly increases the likelihood of triggering naive pruning. From equation 6, we see that if $\pi_{\sigma_t}^{-i}(I) = 0$, it is unnecessary to enter the sub-game tree to calculate $Q_t^i(I, a)$. Similarly, equation 8 shows that if $\pi_{\sigma_t}^i(I) = 0$, it is unnecessary to update the average strategy $\bar{\sigma}_t^i(I)$. This pruning greatly enhances the algorithm's efficiency.

CFVFP employs a simple yet effective approach by using BR strategy instead of regret-matching strategy for next iteration. The convergence of CFVFP can be easily proven. First, FP's convergence to a NE classifies it as a regret minimizer Abernethy *et al.* [2011], which also implies that FP satisfies Blackwell approachability Blackwell [1956]. The CFR framework shows that if an algorithm satisfies Blackwell approachability, its counterfactual regrets will converge to zero Zinkevich *et al.* [2007]. Therefore, CFVFP inherently adheres to Blackwell approachability.

Moreover, we can directly begin with the definition of Blackwell approachability and prove that FP fulfills this criterion. The convergence rate is $\mathcal{O}\left(L|\mathcal{A}|T^{-\frac{1}{2}}\right)$, and the detailed proof can be found in B.1. The pseudocode for the CFVFP algorithm is provided in Appendix C.1.

Since CFVFP follows Blackwell approachability, it is fully compatible with various CFR variants, including MCCFR, CFR+, and different averaging schemes Brown and Sandholm [2019a], resulting in a range of CFVFP variants. Consequently, we conducted a series of experiments to examine the convergence rates of these CFVFP variants. As detailed in Appendix B, the vanilla-weighted MCCFVFP, a combination of MCCFR and CFVFP, shows the fastest convergence among the variants. The pseudocode for the MCCFVFP algorithm is provided in Appendix C.2.

### 4.2 Theoretical Analysis of MCCFVFP Algorithm

We discuss the advantages of the MCCFVFP algorithm in terms of the computing resources required per information set and the algorithm's pruning efficiency.

The MCCFVFP algorithm is highly efficient in conserving computing resources. For example, if an information set has $|\mathcal{A}(I)| = x$ possible actions, MCCFVFP only requires $2x + 1$ additions to process this information set, whereas MCCFR needs $6x - 2$ additions and $3x$ multiplications to complete the same calculation. This means that, for a single information set, MCCFVFP requires only about $2/9$ of the computational time compared to MCCFR. Considering that a Blueprint training typically traverses at least 1 billion nodes, MCCFVFP offers significant advantages over MCCFR in terms of engineering implementation. For a detailed proof, refer to Appendix E.1.

Additionally, CFVFP is highly efficient in game tree pruning. Pruning is a common optimization technique used in all tree search algorithms. For instance, the Alpha-Beta algorithm in complete-information extensive-form games is a pruned version of the Min-Max algorithm, and it was a key factor in Deep Blue's success Hsu [2002]. Naive pruning is the simplest pruning method in CFR. In naive pruning, if no player has a probability of reaching the current state $s$ ($\forall i, \pi_{\sigma_t}^{-i}(s) = 0$), the entire subtree at that state can be pruned for the current iteration without affecting regret calculations.

By analyzing the frequency of different node types in the game tree, we theoretically prove that CFVFP can significantly reduce the number of nodes that need to be processed. Compared to CFR, which must traverse all $\mathcal{O}\left(|\mathcal{S}|\right)$ nodes, CFVFP only needs to traverse $\mathcal{O}\left(\sqrt[|\mathcal{N}|]{|\mathcal{S}|}\right)$ nodes. This is especially beneficial in multiplayer games, where it greatly improves the efficiency of algorithm iterations. For detailed proof, please refer to Appendix E.2.

## 5 Experiments

### 5.1 Description of the Game and Experimental Settings

We employed various game models, such as Kuhn-extension poker, Leduc-extension poker, the princess and monster game Lanctot *et al.* [2010], and Texas Hold'em, to evaluate the performance of different algorithms. These games are widely used benchmarks for comparing algorithmic convergence rates. Kuhn-extension poker, an expanded version of the classic Kuhn poker Kuhn [1950], features an increased card count of $x$, a broader range of betting actions with $y$ options, and up to $z$ raising opportunities. Similarly, Leduc-extension poker is an advanced version of the original Leduc poker Shi and Littman [2002], allowing for flexible scaling. The princess and monster game represents a classic pursuit-evasion problem. Texas Hold'em, one of the most popular poker games in the world, was also included in our analysis.

In Kuhn, Leduc, and princess and monster games, we use a random distribution as the initial strategy to initialize all algorithms. In each iteration, the strategies and regrets of all players are updated simultaneously. In MCCFR, when $R_t^{i,\max} = 0$, the next stage strategy is set to a random pure strategy. In addition, the settings in our comparison experiment are consistent with those in previous groundbreaking works, including MCCFR Lanctot *et al.* [2010], DCFR+ Brown and Sandholm [2019a], and PCFR Farina *et al.* [2021]. For engineering implementation, any node with a probability less than $10^{-20}$ during iteration will be pruned. Our Texas Hold'em experiments employed a variant of the multivalued state technique Brown *et al.* [2018], with an experimental setup closely mirroring that of Pluribus Brown and Sandholm [2019b]. The Texas Hold'em experiments were run on a 32-core, 128GB memory server, while the other experiments were conducted on a single core. It is important to note that all times mentioned below have been converted to reflect execution on a single CPU core.

For a comprehensive exposition of these games and additional experimental findings, please refer to Appendix F. In Appendix G, we compare time, number of iterations, and nodes touched as an indicator. Finally, we use nodes touched and time as an indicator.

### 5.2 Experimental results

As shown in the Figure 2, we can demonstrate the core findings of our paper:

- In small-scale problems like vanilla Kuhn, algorithms such as DCFR, PCFR, and CFR+ may outperform MCCFR and MCCFVFP. However, as the game scale increases, the convergence speed of sampling-based algorithms (MCCFR/MCCFVFP) gradually surpasses that of full-traversal algorithms (DCFR/PCFR/CFR+).

- In our experiments, MCCFVFP consistently converged faster than MCCFR. While theoretically, our algorithm might be less effective than MCCFR in tangled games, the acceleration in its implementation and the tendency of large-scale games to be clear games have led to MCCFVFP outperforming MCCFR in all tested scenarios.

Specifically, when using the number of nodes touched as a metric, MCCFVFP demonstrates a slightly faster convergence rate compared to ES-MCCFR. However, in terms of processing the same number of nodes, MCCFVFP's computation time is only about 2/9 of MCCFR's (the underlying reasons are discussed in Section E.1). As a result, when factoring in the time required for game simulation, MCCFVFP achieves approximately 50% time savings compared to ES-MCCFR for similar levels of exploitability in these games.

As previously noted, algorithms such as CFR+, PCFR, and DCFR are not well-suited for large-scale games due to their reliance on full traversal, which is impractical in complex settings like Texas

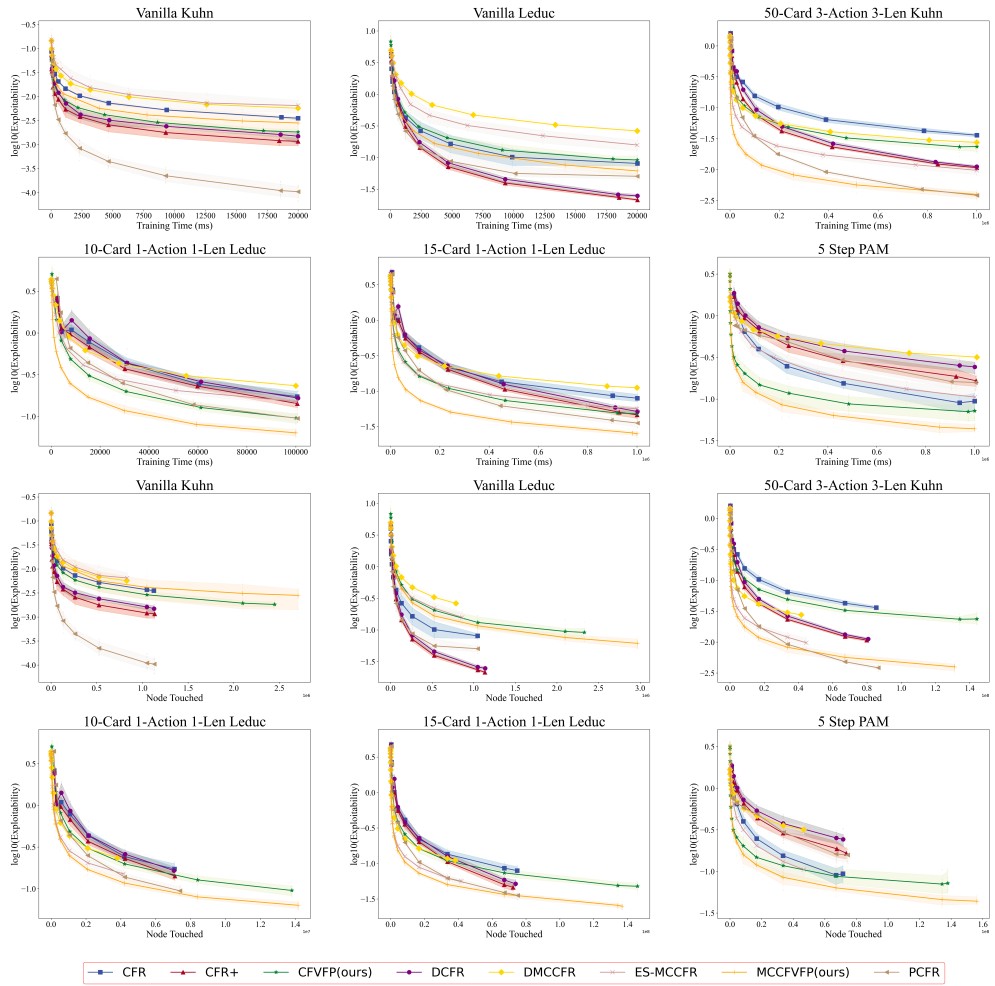

Figure 2: Convergence rates in Kuhn-extension, Leduc-extension, and princess-and-monster games are shown. In the first two rows, time is measured in milliseconds (ms). The last two rows reflect the same running time but with the horizontal axis representing the number of nodes touched during iteration. All experiments tested over an average of 30 rounds. The shaded areas indicate 90% confidence intervals for the trials.

Hold'em. Therefore, in our Texas Hold'em experiments, we focused on comparing the performance of our algorithm specifically against the traditional MCCFR method, excluding these full-traversal algorithms from testing.

In our experiments, we used 2, 3, and 6-player Texas Hold'em game setups, where each player started with 25 Big Blinds (BB) in chips. We assumed that all players would check from the start of the game through to the river stage, simulating a conservative gameplay scenario to examine strategic strengths under limited betting dynamics.

In the two-player Texas Hold'em game, convergence speed can be directly measured using exploitability. The results are shown in Table 1. The public cards in the experiment were 5d2s9d2c7c, and after abstracting the hands, there were 69 hand strength ranks. This sub-game contains approximately 89k information sets, with exploitability measured in BB/100. Compared to MCCFR, MCCFVFP achieved a 20–30% faster convergence speed within the same time frame in the two-player Texas Hold'em game.

| Method | Training time (s) | | | | |
|---|---|---|---|---|---|
| | 7.74 | 23.6 | 70.4 | 730 | 7246 |
| MCCFVFP | 18.8 | 6.15 | 2.68 | 0.776 | 0.289 |
| MCCFR | 25.4 | 9.22 | 3.78 | 0.935 | 0.358 |

Table 1: The convergence rates in two-player Texas Hold'em

| Number of players | Public Cards | Number of hand ranks | Number of information sets | Training time (s) | Number of nodes touched (M) | | Battle result (BB/100) | | |
|---|---|---|---|---|---|---|---|---|---|
| | | | | | MCCFVFP | MCCFR | $r_1$ | $r_2$ | CI |
| 3 | 5d2s9d2c7c | 69 | 63k | 4.3 | 3.3 | 2.7 | 1.390 | -1.440 | ± 0.021 |
| | | | | 40.0 | 30.5 | 25.0 | 0.268 | -0.277 | ± 0.015 |
| | | | | 432.0 | 301.0 | 246.0 | 0.014 | -0.014 | ± 0.003 |
| | KsQsJs3h2h | 124 | 113k | 4.2 | 3.3 | 2.9 | 0.858 | -0.922 | ± 0.026 |
| | | | | 40.0 | 29.9 | 25.6 | 0.182 | -0.201 | ± 0.015 |
| | | | | 410.6 | 287.0 | 244.0 | 0.013 | -0.020 | ± 0.004 |
| 6 | 5d2s9d2c7c | 69 | 188k | 3.6 | 2.1 | 1.7 | 5.380 | -5.950 | ± 0.071 |
| | | | | 35.2 | 18.8 | 12.6 | 0.515 | -0.454 | ± 0.010 |
| | | | | 374.4 | 185.0 | 144.0 | 0.096 | -0.098 | ± 0.002 |
| | KsQsJs3h2h | 124 | 338k | 3.4 | 2.0 | 1.6 | 7.930 | -8.540 | ±0.092 |
| | | | | 32.0 | 17.7 | 13.6 | 0.935 | -0.902 | ± 0.013 |
| | | | | 320.0 | 172.0 | 131.0 | 0.035 | -0.023 | ± 0.004 |

Table 2: The results of different AIs competing against each other in multiplayer games

In multiplayer situations, the exploitability cannot be directly calculated. Therefore, we use the method of mutual battles to measure the strength of different algorithms. As shown in Table 2, in the competition 1, MCCFVFP AI is randomly set as player $i$, and all other players except player $i$ are set as MCCFR AI. Define $r_1$ as the rewards of the MCCFVFP AI player at competition 1. The second setting is exactly the opposite. Player $i$ is randomly set as MCCFR AI, and the remaining players are set as MCCFVFP AI. Define $r_2$ as the rewards of the MCCFR AI player at competition 2. By comparing $r_1$ and $r_2$, we can roughly compare the convergence speeds of different algorithms.

In these scenarios, the MCCFVFP AI, trained for the same duration, significantly outperforms the MCCFR algorithm across all aspects. The 30 to 40 seconds training experiment is particularly important, as it closely mirrors the setup of the Pluribus experiment. In the 6-player Pluribus experiment, the AI trained with MCCFR gained an average of 3.2 BB/100 per game, with a standard error of 1.5 BB/100. In our experiment, using community cards KsQsJs3h2h, MCCFVFP gained an average of 0.932 BB/100 compared to MCCFR—a substantial improvement, especially considering Pluribus gained 3.2 BB/100 against human players.

## 6  Conclusion

This paper introduces a novel method for solving large-scale incomplete information zero-sum games: Monte Carlo Counterfactual Value-Based Fictitious Play (MCCFVFP). By implementing the BR strategy in place of the regret-matching strategy, MCCFVFP achieved convergence speeds approximately 20% to 50% faster than the most advanced MCCFR variants.

In future research, we aim to evaluate the scalability of our method and its compatibility with different CFR variants, including those incorporating deep networks Brown *et al.* [2019]. Furthermore, given MCCFVFP's efficacy in clear games, we envision developing a *warm start* algorithm. This approach would initially use MCCFVFP to eliminate dominated strategies and then switch to algorithms that can further accelerate convergence in the later stages of training.

## Acknowledgements

This work was supported by the National Natural Science Foundation of China (Grant No. 62103158). We would also like to extend our gratitude to Thomas Tellier for his invaluable assistance with our Texas Hold'em experiment. For more information, he can be reached via email at `thomas@gtoking.com` or through his website at gtoking.com.

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

# A   A Toy Example of RM not Performing Well in Clear Game

In training, RM and its variants do not perform well in the clear games, which we attribute to excessive exploration of dominated strategies by RM. For instance, consider the rock-paper-scissors (RPS) game, where an additional action, Leaky Rock (LR), is introduced for player 1. Set the payoff of action LR to be 0.1 less than that of action Rock (R) (making LR a dominated strategy that no rational player would choose):

$$\begin{bmatrix} 0 & -1 & 1 \\ 1 & 0 & -1 \\ -1 & 1 & 0 \end{bmatrix} \xrightarrow{\text{add action LR}} \begin{bmatrix} 0 & -1 & 1 \\ 1 & 0 & -1 \\ -1 & 1 & 0 \\ -0.1 & -1.1 & 0.9 \end{bmatrix} \tag{15}$$

Assuming that player 1 starts with an average strategy and player2 takes scissors in the first round, player 1's payoff is $u^1 = 0.225$. The regret value is $R^1(\text{R}) = 1 - 0.225 = 0.775$, $R^1(\text{S}) = 0 - 0.225 = -0.225$, $R^1(\text{P}) = -1 - 0.225 = -1.225$, $R^1(\text{LR}) = 0.9 - 0.225 = 0.675$, Under the RM algorithm, the probability of selecting Rock in the next iteration becomes: $\sigma^1(\text{R}) \approx 0.534$, and for the Leaky Rock action: $\sigma^1(\text{LR}) \approx 0.466$. Despite LR being a dominated strategy, RM still assigns it nearly half the selection probability, arguably wasting nearly half the computational resources. However, in Fictitious Play (FP), the LR action would never be chosen.

# B   Convergence of CFVFP

## B.1   Blackwell Approachability Game

**Definition 1** *A Blackwell approachability game in normal-form two-player games can be described as a tuple $(\Sigma, u, S^1, S^2)$, where $\Sigma$ is a strategy profile, $u$ is the payoff function, and $S^i = \mathbb{R}^{|\mathcal{A}^i|}_{\leq 0}$ is a closed convex target cone. The Player $i$'s regret vector of the strategy profile $\sigma$ is $R^i(\sigma) \in \mathbb{R}^{|\mathcal{A}^i|}$, for each component $R^i(\sigma)(a_x) = u^i\left(a_x, \sigma^{-i}\right) - u^i\left(\sigma\right), a_x \in \mathcal{A}^i$ the average regret vector for players $i$ to take actions at $T$ time $a$ is $\bar{R}^i_T$*

$$\bar{R}^i_T = \frac{1}{T} \sum_{t=1}^{T} R^i(\sigma_t) \tag{16}$$

*At each time $t$, the two players interact in this order:*

- *Player 1 chooses a strategy $\sigma^1_t \in \Sigma^1$;*

- *Player 2 chooses an action $\sigma^2_t \in \Sigma^2$, which can depend adversarially on all the $\sigma^t$ output so far;*

- *Player 1 gets the vector value payoff $R^1(\sigma_t) \in \mathbb{R}^{l^1}$.*

*The goal of Player 1 is to select actions $\sigma^1_1, \sigma^1_2, \ldots \in \Sigma^1$ such that no matter what actions $\sigma^2_1, \sigma^2_2, \ldots \in \Sigma^2$ played by Player 2, the average payoff vector converges to the target set $S^1$.*

$$\min_{\hat{s} \in S^1} \left\| \hat{s} - \bar{R}^1_T \right\|_2 \to 0 \quad as \quad T \to \infty \tag{17}$$

Before explaining how to choose the action $\sigma_t$ to ensure this goal achieve, we first need to define the forceable half-space:

**Definition 2** *Let $\mathcal{H} \subseteq \mathbb{R}^d$ as half-space, that is, for some $\boldsymbol{a} \in \mathbb{R}^d$, $b \in \mathbb{R}$, $\mathcal{H} = \left\{\boldsymbol{x} \in \mathbb{R}^d : \boldsymbol{a}^\top \boldsymbol{x} \leq b\right\}$. In Blackwell approachability games, the halfspace $\mathcal{H}$ is said to be forceable if there exists a strategy $\sigma^{i*} \in \Sigma^i$ of Player $i$ that guarantees that the regret vector $R^i(\sigma)$ is in $\mathcal{H}$ no matter the strategy played by Player $-i$, such that*

$$R^i\left(\sigma^{i*}, \hat{\sigma}^{-i}\right) \in \mathcal{H} \quad \forall \hat{\sigma}^{-i} \in \Sigma^{-i} \tag{18}$$

*And $\sigma^{i*}$ is forcing action for $\mathcal{H}$.*

Blackwell's approachability theorem states the following.

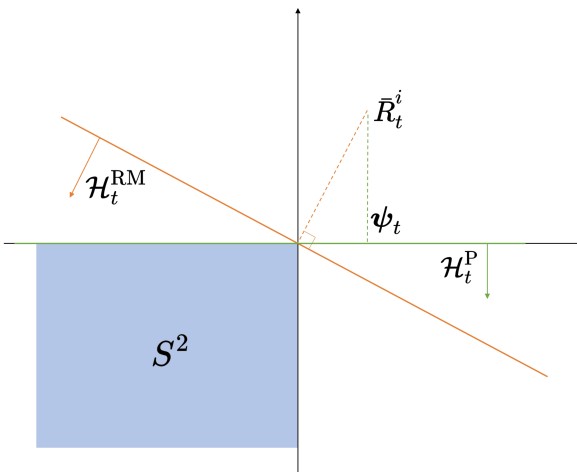

Figure 3: The difference between RM and FP (CFVFP in normal-form game) in a two-dimensional plane

**Theorem 1** *Goal 17 can be attained if and only if every halfspace $\mathcal{H}_t \supseteq S$ is forceable.*

The relationship between Blackwell approachability and no-regret learning is:

**Theorem 2** *Any strategy (algorithm) that achieves Blackwell approachability can be converted into an algorithm that achieves no-regret, and vice versa* Abernethy *et al.* *[2011]*

Let $\bar{\sigma}_T^i$ be the average strategy of player $i$:

$$\bar{\sigma}_T^i(a) = \frac{\sum_{t=1}^T \sigma_t^i(a)}{T} \tag{19}$$

In a two-player zero-sum game, the exploitability of the average strategy $\bar{\sigma}_T^i$ at time $T$ of player $i$ is $\epsilon_T^i = \max_{a' \in \mathcal{A}^i} \bar{R}_T^i(a')$ Brown [2020]. Obviously, the regret value is always greater than the exploitability:

$$\lim_{T \to \infty} \epsilon_T = \lim_{T \to \infty} \sum_{i \in \mathcal{N}} \epsilon_T^i \leq \lim_{T \to \infty} \sum_{i \in \mathcal{N}} \min_{\hat{s} \in S^i} \left\| \hat{\boldsymbol{s}} - R_T^i \right\|_2 = 0 \tag{20}$$

So, if the algorithm achieves Blackwell approachability, the average strategy $\bar{\sigma}_T^i$ will converge to equilibrium with $T \to \infty$. The rate of convergence is $\epsilon_T^i \leq \bar{R}_T^i \leq L\sqrt{|\mathcal{A}|}/\sqrt{T}$.

### B.2 Fictitious Play achieves Blackwell Approachability

We first prove that FP achieves Blackwell approachability in a two-player zero-sum game. Define $\bar{R}_t^{i,\max}$ be the maximum portion of vector $\bar{R}_t^{i,+}$. If $\bar{R}_t^{i,\max} \neq \boldsymbol{0}$, we find the point $\boldsymbol{\psi}_t = \bar{R}_t^i - \bar{R}_t^{i,\max}$ in $\mathbb{R}^{|\mathcal{A}^i|}$ and let $\frac{\bar{R}_t^i - \boldsymbol{\psi}_t}{|\bar{R}_t^i - \boldsymbol{\psi}_t|}$ be the normal vector, then we can determine half-space by $\frac{\bar{R}_t^i - \boldsymbol{\psi}_t}{|\bar{R}_t^i - \boldsymbol{\psi}_t|}$ and $\boldsymbol{\psi}_t$:

$$\mathcal{H}_t^{\mathrm{P}} = \left\{ \boldsymbol{z} \in \mathbb{R}^{l^{-i}} : (\bar{R}_t^i - \boldsymbol{\psi}_t)^\top \boldsymbol{z} \leq (\bar{R}_t^i - \boldsymbol{\psi}_t)^\top \boldsymbol{\psi}_t \right\} \tag{21}$$

Because $\bar{R}_t^i - \boldsymbol{\psi}_t = \bar{R}_t^{i,\max}$, $(\bar{R}_t^i - \boldsymbol{\psi}_t)^\top \boldsymbol{\psi}_t = 0$, therefore:

$$\mathcal{H}_t^{\mathrm{P}} = \left\{ \boldsymbol{z} \in \mathbb{R}^{l^{-i}} : \left\langle \bar{R}_t^{i,\max}, \boldsymbol{z} \right\rangle \leq 0 \right\} \tag{22}$$

For any point $\boldsymbol{s}' \in S^i$ there is $\left\langle \bar{R}_t^{i,\max}, \boldsymbol{s}' \right\rangle \leq 0$. Then we need to prove that a forcing action for $\mathcal{H}_t^{\mathrm{P}}$ indeed exists. According to Definition 2, we need to find a $\sigma_{t+1}^{i*} \in \Sigma^i$ that achieves $R^i\left(\sigma_{t+1}^{i*}, \hat{\sigma}_{t+1}^{-i}\right) \in \mathcal{H}_{t+1}^{i,\mathrm{P}}$ for any $\hat{\sigma}_{t+1}^{-i} \in \Sigma^{-i}$. For simplicity, let $\boldsymbol{\ell} = [u^i\left(a_1, \sigma^{-i}\right), \dots]^\top \in \mathbb{R}^{|\mathcal{A}^i|}$,

we rewrite the regret vector as $R^i \left( \sigma_{t+1}^{i*}, \hat{\sigma}_{t+1}^{-i} \right) = \boldsymbol{\ell} - \left\langle \boldsymbol{\ell}, \sigma_{t+1}^{i*} \right\rangle \mathbf{1}$, we are looking for a $\sigma_{t+1}^{i*} \in \Sigma^i$ such that:

$$
\begin{aligned}
& R^i \left( \sigma_{t+1}^{i*}, \hat{\sigma}_{t+1}^{-i} \right) \in \mathcal{H}_t^{\mathrm{P}} \\
\Longleftrightarrow & \left\langle \bar{R}_t^{i,\mathrm{max}}, \boldsymbol{\ell} - \left\langle \boldsymbol{\ell}, \sigma_{t+1}^{i*} \right\rangle \mathbf{1} \right\rangle \le 0 \\
\Longleftrightarrow & \left\langle \bar{R}_t^{i,\mathrm{max}}, \boldsymbol{\ell} \right\rangle - \left\langle \boldsymbol{\ell}, \sigma_{t+1}^{i*} \right\rangle \left\langle \bar{R}_t^{i,\mathrm{max}}, \mathbf{1} \right\rangle \le 0 \\
\Longleftrightarrow & \left\langle \bar{R}_t^{i,\mathrm{max}}, \boldsymbol{\ell} \right\rangle - \left\langle \boldsymbol{\ell}, \sigma_{t+1}^{i*} \right\rangle \left\| \bar{R}_t^{i,\mathrm{max}} \right\|_1 \le 0 \\
\Longleftrightarrow & \left\langle \boldsymbol{\ell}, \frac{\bar{R}_t^{i,\mathrm{max}}}{\left\| \bar{R}_t^{i,\mathrm{max}} \right\|_1} \right\rangle - \left\langle \boldsymbol{\ell}, \sigma_{t+1}^{i*} \right\rangle \le 0 \\
\Longleftrightarrow & \left\langle \boldsymbol{\ell}, \frac{\left[ \bar{R}_t^i \right]^{\mathrm{max}}}{\left\| \left[ \bar{R}_t^i \right]^{\mathrm{max}} \right\|_1} - \sigma_{t+1}^{i*} \right\rangle \le 0
\end{aligned}
\tag{23}
$$

We obtain the strategy $\sigma_{t+1}^{i*} = \frac{\bar{R}_t^{i,\mathrm{max}}}{\left\| \bar{R}_t^{i,\mathrm{max}} \right\|_1}$ that guarantees $\mathcal{H}_{t+1}^{\mathrm{P}}$ to be forceable half space. And the action with the highest regret value is actually the BR strategy Brown [2020], so FP achieves Blackwell approachability. Figure 3 shows the difference in forceable half spaces for BR and RM strategies in the two-dimensional plane. According to 2, BR strategy is also a regret minimizer in normal-form game. Therefore, replacing the RM strategy with the BR strategy in CFR does not affect the convergence of the algorithm.

### B.3 The convergence speed of FP and clear games

However, it is important to note that the convergence point of FP differs from that of RM . Specifically, the convergence point for FP is $\psi_t = \bar{R}_t^i - \bar{R}_t^{i,\mathrm{max}}$, while for RM it is $\psi_t^{\mathrm{RM}} = \bar{R}_t^i - \bar{R}_t^{i,+} = \bar{R}_t^{-i}$. The relationship between these two points can be expressed as $\|\psi_t^{\mathrm{RM}}\|_2 \le \|\psi_t\|_2 \sqrt{|\mathcal{A}|}$. The convergence rate of RM is $\mathcal{O}\left( L \sqrt{|\mathcal{A}|/T} \right)$. Consequently, the overall convergence rate for FP is $\mathcal{O}\left( L|\mathcal{A}|/\sqrt{T} \right)$.

In Section 3, we have analyzed that the RM strategy may select a dominated strategy, whereas FP will not. Therefore, representing the complexity of the problem using $|\mathcal{A}|$ for FP is inaccurate. If an action $a'$ is a dominated strategy, then it cannot be the best response, i.e., $b(\sigma) \ne a'$. Hence, in FP, we can replace $|\mathcal{A}|$ with $\mathcal{A}$nd, where $A_{\mathrm{nd}}^i \subseteq \mathcal{A}^i$ is the set of non-dominated strategies in the game. This adjustment leads to the FP convergence rate being $\mathcal{O}\left( L|\mathcal{A}_{\mathrm{nd}}|/\sqrt{T} \right)$.

We now define a "clear game" as follows: when the number of non-dominated strategies in a game satisfies $|\mathcal{A}_{\mathrm{nd}}| \le \sqrt{|\mathcal{A}|}$, the game is considered a clear game. In Figure 1, the square root of 100 is 10, aligning with theoretical expectations. In clear games, even when the number of iterations is used as a measure, the convergence speed of FP will outperform that of RM.

## C Pseudocode

### C.1 CFVFP

The pseudocode 1 here does not join the rest of the variants, nor does it consider cases other than two-player zero-sum.

### C.2 MCCFVFP

Since MCCFVFP will directly sample actions on opportunity nodes, the efficiency of the MCCFVFP algorithm will be higher than CFVFP. The pseudocode of MCCFVFP is shown in pseudocode 2.

**Algorithm 1** CFVFP($s, \pi, \pi^c$)

1: **if** $\pi^1 = \pi^2 = 0$ or $\pi^c = 0$ **then**
2:     **return** $[0, 0]$
3: **end if**
4: **if** $s \in \mathcal{Z}$ **then**
5:     **return** $\left[u^1(s)\pi^2\pi^c, u^2(s)\pi^1\pi^c\right]$
6: **end if**
7: Define $I$ as the information set to which $s$ belongs
8: $r = [0, 0]$
9: **if** $P(s) = c$ **then**
10:     **for** $a \in \mathcal{A}(s)$ **do**
11:         $r' =$PCFR($s + a, \pi, \pi^c\sigma^c(s)(a)$)
12:         $r = r + r'(p_s)$
13:     **end for**
14:     **return** r
15: **else**
16:     **if** $P(s) = p_1$ **then**
17:         $\pi_s, \pi_o = \pi^1, \pi^2$
18:         $p_s, p_o = p_1, p_2$
19:     **else**
20:         $\pi_s, \pi_o = \pi^2, \pi^1$
21:         $p_s, p_o = p_2, p_1$
22:     **end if**
23:     **if** $\pi_o = 0$ **then**
24:         $r' = $CFVFP($s + \sigma_t(I), \pi, \pi^c$)
25:         $r(p_o) = r'(p_o)$
26:         $\bar{\sigma}_t^{p_s}(I) = ((t-1)\bar{\sigma}_{t-1}^{p_s}(I) + \sigma_t^{p_s}(I))/t$
27:     **else if** $\pi_s = 0$ **then**
28:         **for** $a \in \mathcal{A}(I)$ **do**
29:             $r' = $CFVFP($s + a, \pi, \pi^c$)
30:             $Q_t^{p_s}(I, a) = Q_{t-1}^{p_s}(I, a) + r'(p_s)$
31:             **if** $a = \sigma^{p_s}(I)$ **then**
32:                 $r(p_o) = r'(p_o)$
33:             **end if**
34:         **end for**
35:         $\sigma_{t+1}^{p_s}(I) = \max_{a \in \Sigma_p^i(I)} Q_t^{p_s}(I, a)$
36:     **else**
37:         **for** $a \in \mathcal{A}(I)$ **do**
38:             $\pi^{p_s} = \sigma_t^{p_s}(I, a)$
39:             $r' = $CFVFP($s + a, \pi, \pi^c$)
40:             $Q_t^{p_s}(I, a) = Q_{t-1}^{p_s}(I, a) + r'(p_s)$
41:             $r(p_o) = r(p_o) + r'(p_o)$
42:             $r(p_s) = r(p_s) + r'(p_s)\sigma_t^{p_s}(I, a)$
43:         **end for**
44:         $\bar{\sigma}_t^{p_s}(I) = ((t-1)\bar{\sigma}_{t-1}^{p_s}(I) + \sigma_t^{p_s}(I))/t$
45:         $\sigma_{t+1}^{p_s}(I) = \max_{a \in \Sigma_p^i(I)(I)} Q_t^{p_s}(I, a)$
46:     **end if**
47: **end if**
48: **return** r

# D   Comparison between variants of CFVFP

## D.1   CFVFP variants

We integrated CFVFP with RM+ and MC variants, resulting in four distinct combinations: CFVFP, CFVFP+, MCCFVFP, and MCCFVFP+. Figure 4 clearly illustrates that MCCFVFP and MCCFVFP+ consistently outperform CFVFP and CFVFP+ in terms of results. However, the performance of

**Algorithm 2** MCCFVFP$(s, \pi)$

---

1: **if** $\pi^1 = \pi^2 = 0$ **then**
2:     **return** $[0, 0]$
3: **end if**
4: **if** $s \in \mathcal{Z}$ **then**
5:     **return** $\left[ u^1(s)\pi^2, u^2(s)\pi^1 \right]$
6: **end if**
7: Define $I$ as the information set to which $s$ belongs
8: $r = [0, 0]$
9: **if** $P(s) = c$ **then**
10:     $a \sim \sigma^c(s)$
11:     $r =$ PMCCFR$(s + a, \pi)$
12:     **return** r
13: **else**
14:     **if** $P(s) = p_1$ **then**
15:         $\pi_s, \pi_o = \pi^1, \pi^2$
16:         $p_s, p_o = p_1, p_2$
17:     **else**
18:         $\pi_s, \pi_o = \pi^2, \pi^1$
19:         $p_s, p_o = p_2, p_1$
20:     **end if**
21:     **if** $\pi_o = 0$ **then**
22:         $r' = $ MCCFVFP$(s + \sigma_t(I), \pi)$
23:         $r(p_o) = r'(p_o)$
24:         $\bar{\sigma}_t^{p_s}(I) = ((t-1)\bar{\sigma}_{t-1}^{p_s}(I) + \sigma_t^{p_s}(I))/t$
25:     **else if** $\pi_s = 0$ **then**
26:         **for** $a \in \mathcal{A}(I)$ **do**
27:             $r' = $ MCCFVFP$(s + a, \pi)$
28:             $Q_t^{p_s}(I, a) = Q_{t-1}^{p_s}(I, a) + r'(p_s)$
29:             **if** $a = \sigma^{p_s}(I)$ **then**
30:                 $r(p_o) = r'(p_o)$
31:             **end if**
32:         **end for**
33:         $\sigma_{t+1}^{p_s}(I) = \max_{a_{\mathrm{P}} \in \Sigma_{\mathrm{P}}(I)} Q_t^{p_s}(I, a)$
34:     **else**
35:         **for** $a \in \mathcal{A}(I)$ **do**
36:             $\pi^{p_s} = \sigma_t^{p_s}(I, a)$
37:             $r' = $ MCCFVFP$(s + a, \pi)$
38:             $Q_t^{p_s}(I, a) = Q_{t-1}^{p_s}(I, a) + r'(p_s)$
39:             $r(p_o) = r(p_o) + r'(p_o)$
40:             $r(p_s) = r(p_s) + r'(p_s)\sigma_t^{p_s}(I, a)$
41:         **end for**
42:         $\bar{\sigma}_t^{p_s}(I) = ((t-1)\bar{\sigma}_{t-1}^{p_s}(I) + \sigma_t^{p_s}(I))/t$
43:         $\sigma_{t+1}^{p_s}(I) = \arg\max_{a_{\mathrm{P}} \in \Sigma_{\mathrm{P}}(I)} Q_t^{p_s}(I, a)$
44:     **end if**
45: **end if**
46: **return** r

---

MCCFVFP and MCCFVFP+ varies across different problem sets. Based on these observations, we ultimately selected the more conservative MCCFVFP method.

### D.2 Weighted Averaging Schemes for CFVFP

Different weights of CFVFP will also have a significant impact on the results. The weight $w_t$ is introduced into $Q_t^i(I)$:

$$Q_t^i(I) = \begin{cases} Q_{t-1}^i(I) + w_t u^i(I, \sigma_t) & \text{if } \pi_{\sigma_t}^{-i}(I) = 1 \\ Q_{t-1}^i(I) & \text{otherwise.} \end{cases} \tag{24}$$

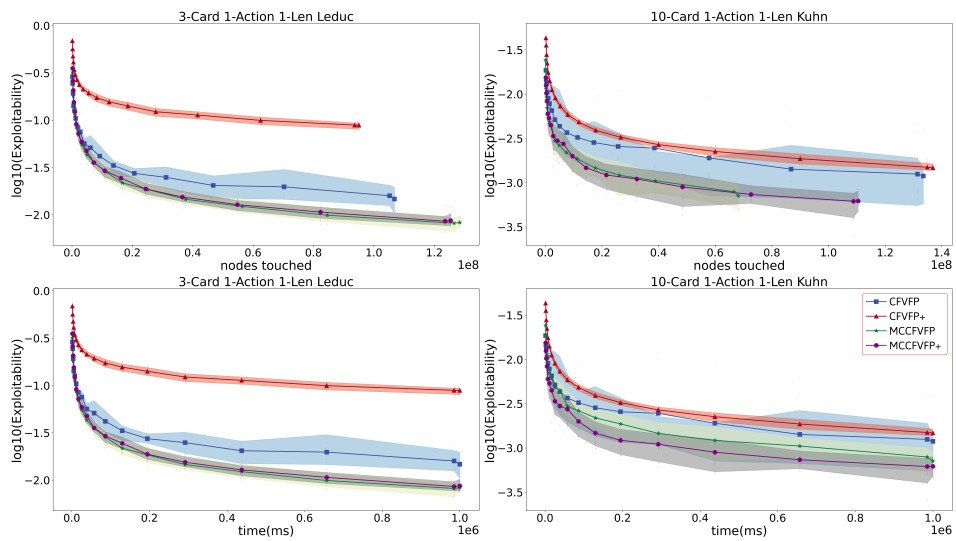

Figure 4: Convergence rate of MCCFVFP variants in different games

Common $w_t$ values are $w_t = 1$, $w_t = \log t$, $w_t = t$, $w_t = t^2$. The Figures 5 show the experimental results with different weights. In the comparison of four common $w_t$, convergence is faster when

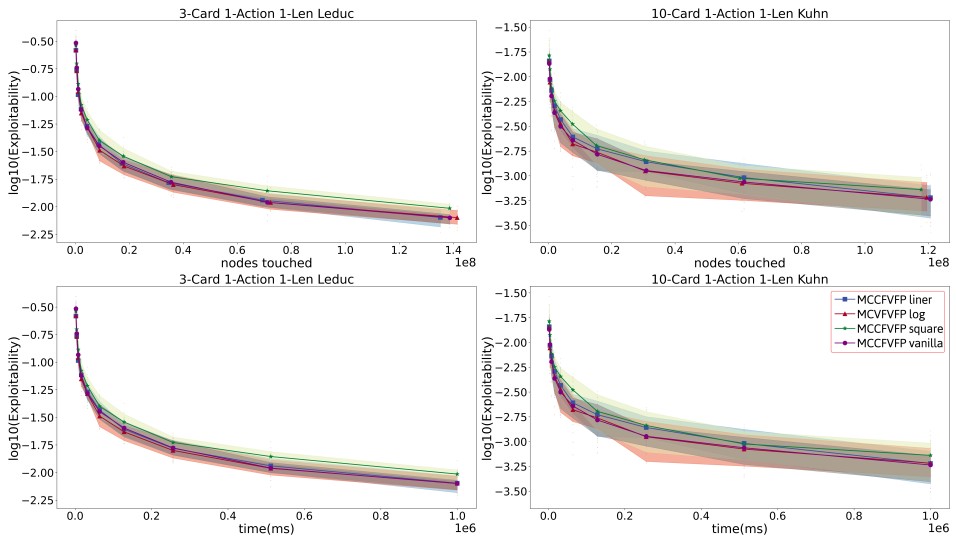

Figure 5: Convergence rate of different weighted average schemes for CFVFP

$w_t = 1$.

# E    Proof of conclusion in theoretical analysis of CFVFP algorithm

## E.1    Time Complexity of CFR and CFVFP in an Information Set

Due to its significantly streamlined calculation process, CFVFP requires far less computation per information set compared to CFR . Consider an information set $I$ with $x$ possible actions, denoted as $\mathcal{A}(I) = a_1, a_2, \ldots, a_x$. If the counterfactual payoff $u(I, a)$ for each action is known, then:

**The calculations required for CFVFP in one information set are:**

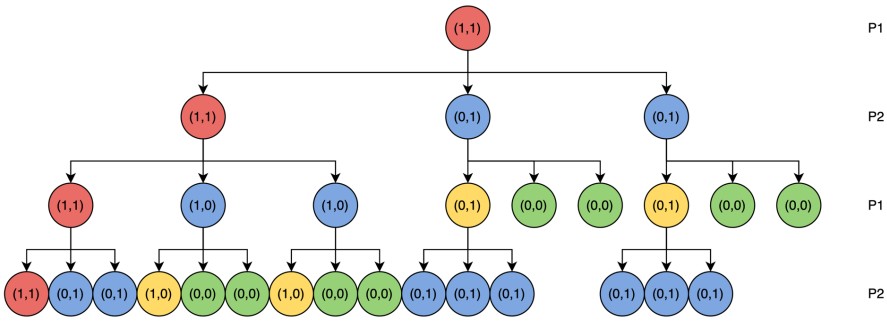

Figure 6: Game tree when each node has $h = 4$ levels and $g = 3$ actions. The number $(\pi^1(s), \pi^2(s))$ in each node represents the probability of player1 and player2 reaching this node respectively.

1. Add the counterfactual payoff $u^i(I, \sigma_t|_{I \to a})$ to the counterfactual value $Q_T^i(I, a)$, this step requires $x$ additions;

2. Find the maximum value of $Q_T^i(I)$ (get BR strategy), this step requires $x$ comparisons;

3. Update the BR strategy to the average strategy, this step requires 1 addition;

A total of $2x + 1$ additions.

**The calculations required for CFR in one information set are:**

1. Get average counterfactual payoff $u^i(I)$ from $u^i(I, \sigma_t|_{I \to a})$ and strategy $\sigma_t(I)$, this step needs $x$ multiplications and $x - 1$ additions;

2. Get regret value $R(I, a)$, this step needs $x$ additions;

3. Multiply the regret value $R(I, a)$ by $\pi_{\sigma_t}^{-i}(I)$ and add it to the average regret value $\bar{R}_T(I, a)$, this step needs $x$ additions and $x$ multiplications;

4. Compared Regret value with 0, this step needs $x$ comparisons;

5. Add up the positive regret values, this step needs $x - 1$ additions;

6. Get regret matching strategy, this step $x$ multiplications;

7. Update the RM strategy to the average strategy, this step needs $x$ additions;

A total of $6x - 2$ additions and $3x$ multiplications. It can be seen that in one information set, CFVFP only takes 2/9 of the time of CFR .

### E.2 Number of nodes touched in one iteration

In a deterministic game tree with $h$ levels and $g$ actions per node (no chance nodes), such as a $g = 3, h = 4$ scenario depicted in Figure 6, nodes are categorized into four distinct types $(s_{\text{all}} = (s_{\text{r}}, s_{\text{g}}, s_{\text{b}}, s_{\text{y}}))$.

- The probability of all players reaching this node is 1 (red node $s_{\text{r}}$);

- The probability of all players reaching this node is 0 (green node $s_{\text{g}}$), and these nodes can be pruned;

- The probability of the current player reaching this node is 1, and the probability of other players is 0 (blue node $s_{\text{b}}$);

- The probability that the current player reaches this node is 0, and the probability of other players is 1 (yellow node $s_{\text{y}}$);

We define the nodes that must be touched as $s_{\text{pass}=(s_{\text{r}}, s_{\text{b}}, s_{\text{y}})}$. The figure shows that each level has a single red node, with yellow and green nodes following a blue node, which is derived from red and

yellow nodes. The function $F_h(s_x)$ quantifies the count of each node type across layers 1 to $h$.

$$
\begin{aligned}
F_h\left(s_{\mathrm{r}}\right) &= 1 \\
F_h\left(s_{\mathrm{g}}\right) &= g F_{h-1}\left(s_{\mathrm{g}}\right) + (g-1) F_{h-1}\left(s_{\mathrm{b}}\right) \\
F_h\left(s_{\mathrm{b}}\right) &= (g-1) F_{h-1}\left(s_{\mathrm{r}}\right) + g F_{h-1}\left(s_{\mathrm{y}}\right) \\
F_h\left(s_{\mathrm{y}}\right) &= F_{h-1}\left(s_{\mathrm{b}}\right)
\end{aligned}
\tag{25}
$$

the general term formula for blue node $s_{\mathrm{b}}$ in the $h$ layer is:

$$
\begin{aligned}
F_h\left(s_{\mathrm{b}}\right) &= (g-1) F_{h-1}\left(s_{\mathrm{r}}\right) + g F_{h-1}\left(s_{\mathrm{y}}\right) \\
&= (g-1)1 + g F_{h-2}\left(s_{\mathrm{b}}\right) \\
&= (g-1) + g\left((g-1) + g F_{h-4}\left(s_{\mathrm{b}}\right)\right) \\
&= \dots
\end{aligned}
\tag{26}
$$

after simplification:

$$
F_{h\geq 2}(s_{\mathrm{b}}) = (g-1) \sum_{i=0}^{\lfloor (h-2)/2 \rfloor} g^i.
\tag{27}
$$

The number of nodes that need to be touched $s_{\mathrm{pass}}$ in the $h$ level game tree is:

$$
\begin{aligned}
F_h\left(s_{\mathrm{pass}}\right) &= F_h\left(s_{\mathrm{r}}\right) + F_h\left(s_{\mathrm{b}}\right) + F_h\left(s_{\mathrm{y}}\right) \\
&= 1 + F_h\left(s_{\mathrm{b}}\right) + F_{h-1}\left(s_{\mathrm{b}}\right)
\end{aligned}
\tag{28}
$$

$$
F_{h\geq 3}(s_{\mathrm{pass}}) = 1 + (g-1) \sum_{i=0}^{\lfloor (h-2)/2 \rfloor} g^i + (g-1) \sum_{i=0}^{\lfloor (h-3)/2 \rfloor} g^i.
\tag{29}
$$

The number of node $s_{\mathrm{all}}$ of the entire $h$ level game tree is:

$$
F_h(s_{\mathrm{all}}) = \sum_{i=0}^{h} g^i = \frac{1-g^h}{1-g}.
\tag{30}
$$

In a two-player game, it can be approximately considered that $F_h(s_{\mathrm{all}}) \propto \sqrt{F_h(s_{\mathrm{pass}})}$, so CFVFP only needs to touch $\mathcal{O}\left(\sqrt{|\mathcal{S}|}\right)$ nodes in one iteration. Extending the results to multi-player games, CFVFP only needs to touch $\mathcal{O}\left(\sqrt[|\mathcal{N}|]{|\mathcal{S}|}\right)$ nodes. At the same time, CFR must traverse all $\mathcal{O}\left(|\mathcal{S}|\right)$ nodes.

## F   Experiment Supplementary Notes

### F.1   Description of the game

#### F.1.1   Kuhn-extension/Leduc-extension poker

We have made improvements to Kuhn and Leduc Poker:

1. The original Leduc and Kuhn poker types are 3 cards, and in the improved game the types are $x \geq 3$ cards;

2. The original Leduc and Kuhn can only raise one fixed chip, but in the improved game it is allowed to raise $y \geq 1$ chips of various sizes. The Bet Action raise size here adopts an equal proportional sequence in multiples of 2. For example, when the blind bet is 1 and $y = 4$, the allowed raise action is [1, 2, 4, 8];

3. The original Leduc and Kuhn can only raise once in a round. After one player raises, the rest of the players can only choose to call or fold, and cannot raise a larger bet. In the improved game, it can be raised up to z times.

**F.1.2   Princess and Monster**

Princess and monster (PAM) is a game in which two players chase and escape in a dungeon with obstacles (a 4-connected grid diagram of $m \times n$). The game rule is:

- The two players are monster and princess, and each player knows the structure of the dungeon and which rooms can be accessed.
- They can only exist in one particular dungeon at a time, and they all know the number of their current dungeon (grid).
- The monster's goal is to find and capture the princess as soon as possible.
- The princess's goal is to escape the monster as much as possible.

The actions they can choose are:

- In the initial stage, monster and princess can choose any birth room to start the game;
- In the following process, monster and princess can choose to move one space per step in four directions, up, down, left and right, or stay in place. When moving, they cannot exceed the dungeon boundary or enter impassable rooms.

Result of the game:

- At the same time, if the monster and princess appear in the same dungeon, the game ends. The princess survives the $n$ round and gets $n$ utilities, and the corresponding monster gets $-n$ utilities.
- If the monster does not capture the princess within a certain number of steps (such as $N$ move), the game ends directly. The princess earns $N$ utilities and the monster earns $-N$ utilities.

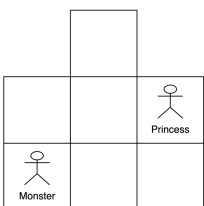

Figure 7:   The structure of the dungeon.

As shown in Figure 7, we set the dungeon to a 3×3 grid, with inaccessible rooms in the upper left and right corners.

**F.2   The Rest of the Experimental Results**

We measured the number of information sets and the number of nodes in different games in Table 3, and there are rest of the experimental results in Figure 8.

# G   The difference between time, nodes touched and iterations as indicator

In previous papers, many convergence analyses were performed based on the number of iterations $T$, which is obviously unfair to the sampling CFR/CFVFP algorithm. For sampling algorithms, many studies have used the number of nodes to analyze the convergence of the algorithm, but these studies have ignored the inconsistent calculation time of different nodes and the inconsistent number of information sets that have passed through the same node. The cost of an iteration in CFR/CFVFP is divided into two parts: the game tree traversal cost and the RM/BR strategy cost calculated on the information set:

- On each information set: It is necessary to calculate the RM/BR strategy and update the average strategy for the passed information set. These costs have been detailed in Section E.1.

| Game name | Information set | Number of nodes |
|---|---:|---:|
| RPS | 2 | 13 |
| 3 C 1 P 1 L Kuhn | 12 | 55 |
| 15 C 1 P 1 L Kuhn | 60 | 1891 |
| 50 C 1 P 1 L Kuhn | 200 | 22051 |
| 3 C 3 P 3 L Kuhn | 48 | 271 |
| 7 C 3 P 3 L Kuhn | 112 | 1891 |
| 7 C 5 P 3 L Kuhn | 364 | 6427 |
| 15 C 5 P 3 L Kuhn | 780 | 32131 |
| 15 C 7 P 3 L Kuhn | 1920 | 80011 |
| 3 C Leduc | 288 | 1945 |
| 7 C Leduc | 1512 | 25985 |
| 15 C Leduc | 6480 | 255601 |
| 25 C Leduc | 18900 | 1180001 |
| 3 C 3 P Leduc | 1680 | 12529 |
| 7 C 3 P Leduc | 9016 | 172313 |
| 15 C 3 P Leduc | 41160 | 1712521 |
| 3 C 5 P Leduc | 6399 | 49384 |
| 7 C 5 P Leduc | 34587 | 685280 |
| 15 C 5 P Leduc | 158355 | 6831856 |
| 4 round PAM | 224 | 68815 |
| 5 round PAM | 794 | 715655 |
| 6 round PAM | 2804 | 7447021 |

Table 3: Information sets and node number record for different games

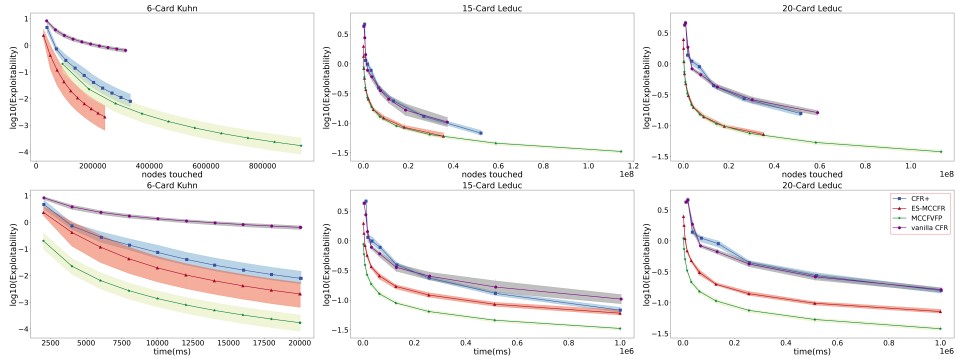

Figure 8: Convergence rate in Leduc-extension, Kuhn-extension, Here Action and Len are both 1.

- There are three types of nodes on each game tree node, all of which need to get the counterfactual utility of each player. The difference is:
  - On the player node, the counterfactual utility needs to be obtained on this type of node $u^i\left(s, \sigma_t|_{I \to a}\right)$ and multiplied by the strategy return.
  - On opportunity nodes, in this type of node need to get virtual utility $u^i\left(I, \sigma_t|_{I \to a}\right)$ and multiplied by opportunity node probability.
  - On the end point node, it is necessary to solve the final income according to the node on this type of node.

When passing through the same number of nodes, different algorithms do not pass through the same number of information sets. For example, when using the CFR algorithm to train vanilla Kuhn poker (assuming no pruning), one iteration passes through 55 nodes (24 player nodes, 1 random node, 30

leaf nodes), and calculate regret matching strategies on 12 information sets, at this time, the ratio of information set calculation and node calculation is 12:55. While using MCCFR the calculation passes through 10 nodes (4 player nodes, 1 random node, 5 leaf nodes), and calculate regret matching strategies on 4 information sets, the ratio of information set calculation and node calculation is 4:10. And the computing time is different at different decision, opportunity, and end point nodes. Therefore, measuring the performance of the algorithm by the number of nodes passed does not objectively reflect the ability of the algorithm.

