# OpenReview forum: "Accelerating Nash Equilibrium Convergence in Monte Carlo Settings Through Counterfactual Value Based Fictitious Play"
_NeurIPS.cc/2024/Conference — NeurIPS 2024 poster_

### Official Review · Reviewer_APT4 · 2024-06-27

**Soundness:** 3
**Presentation:** 3
**Contribution:** 3
**Rating:** 7
**Confidence:** 2

**Summary:**

The paper introduces a novel algorithm, Monte Carlo Counterfactual Value-Based Fictitious Play (MCCFVFP). This algorithm aims to accelerate the convergence of Nash Equilibria in extensive-form imperfect information games. MCCFVFP combines the counterfactual value calculations from Counterfactual Regret Minimization (CFR) with the best response strategy from Fictitious Play (FP). The authors claim that MCCFVFP achieves up to three times faster convergence compared to advanced MCCFR variants and significantly outperforms them in large-scale games with a high proportion of dominated strategies.

**Strengths:**

- The paper introduces a unique and novel combination of counterfactual value calculations with fictitious play, leveraging strengths from both methods.
- The theoretical analysis proving the convergence of MCCFVFP adds a strong foundation to the empirical results.
- MCCFVFP is shown to achieve significantly faster convergence rates in extensive-form games, particularly in scenarios with a high proportion of dominated strategies.

**Weaknesses:**

The paper could provide a more detailed comparison with a broader range of existing algorithms beyond MCCFR variants to establish a more comprehensive performance baseline.

**Questions:**

Can the authors describe a bit about the comparison between MCCFVFP with reinforcement learning approaches for extensive form imperfect information games?

**Limitations:**

yes

---

> ### Author Rebuttal · Authors · 2024-08-05
>
> We thank the reviewer for the comments and observations on our work. Below we address the questions raised and the discussed weaknesses:
>
> ## Provide a more detailed comparison:
>
> We update the experimental evaluation part (in global response part 3 'Experiments'), we compared our algorithm with different algorithms including DCFR and PCFR, both of which are classified as CFR rather than variants of MCCFR. I'm not sure if you want me to compare the algorithm with some reinforcement learning (RL) methods. However, generally, these methods do not perform as well as CFR, the no-regret learning algorithms [1], in imperfect information games, and there are significant differences in their usage. Therefore, we do not consider comparing our algorithm with RL algorithms for now.
>
> ## Describe about the comparison between MCCFVFP with RL approaches:
>
> This is a very insightful viewpoint and also a future research direction for us. Firstly, the purposes of RL and game learning algorithms are different. RL algorithms generally require the environment to provide a reward function, and the goal of RL algorithms is to find a strategy to obtain more rewards. In game learning algorithms, the environment does not provide a fixed reward function, and the goal of game algorithms is to learn a Nash equilibrium strategy, while the Nash equilibrium strategy is often not the case with the most rewards (such as the famous Prisoner's Dilemma).
> However, the algorithm proposed in this paper has many similarities to the Q-learning algorithm in RL at the implementation level, including the definition of Q-values and the update method of the Q-value table. So, how to modify our algorithm(MCCFVFP) to make it conform to the current mainstream RL framework and be used to solve a wider range of problems is an interesting research direction. We will also continue to work on this in the future.
>
> [1] Brown N, Sandholm T. Superhuman AI for multiplayer poker[J]. Science, 2019, 365(6456): 885-890.

---

> > ### Author Response · Authors · 2024-08-12
> >
> > If you have any new questions or still have doubts about our current answers, please feel free to communicate with us at any time. We will try our best to answer any questions you may have.

---

> > > ### Comment · Reviewer_APT4 · 2024-08-13
> > >
> > > Thank the authors for the rebuttal. I have no further questions.

---

### Official Review · Reviewer_KPRH · 2024-07-11

**Soundness:** 3
**Presentation:** 2
**Contribution:** 3
**Rating:** 6
**Confidence:** 3

**Summary:**

The paper proposes a new algorithm for solving extensive-form imperfect information games that relies on Monte Carlo (MC) simulations. The method abbreviated MCCFVFP combines MC settings with Counterfactual Regret Minimization (CFR) and the best response strategy of fictious play. Experimental evaluation demonstrate its speed advantage over the state-of-the-art competitors, in particular in clear games (i.e. games in which the vast majority of strategies are dominated ones).

**Strengths:**

The method clearly outperforms competitive approaches in terms of convergence speed. The method is particularly well suited for clear games, i.e. the ones with over 90% of dominated strategies, as reported by the authors.

**Weaknesses:**

1. The paper has not been carefully revised before submission and there are certain issues that hinder is smooth reading and understanding. For instance, $u^i$ is not explicitly defined in the paper (I assume it is the payoff of player $i$) and in some cases is a one-argument function, in other case a two-argument one (cf. eq. 2). Similarly, $R_{T}^{i,+}$ in eq. 5.

2. A discussion in section 5.2.1. is rather accidental. Some plots in Fig. 2 are addressed while the other are not mentioned (e.g. Princess and Monster plots).

3. RM is not explicitly defined in the paper – it is only defined in the appendix

**Questions:**

-- See points 1 and 2 in weaknesses
-- The results for tangeled games are not so strong as for the clear games. At the same time tangeled games are intuitively much more complex than clear games, which shows certain limitations of the proposed method. Please comment on that.

-- Random games with 21845 nodes do not seem to be challenging. Would the conclusions hold for much larger random games?
-- In section 5.3. the numbers of stored information sets of both methods are equal. Is it really the case or there is a mistake there?

**Limitations:**

The limitations are not explicitly specified.

---

> ### Author Rebuttal · Authors · 2024-08-06
>
> We are extremely thankful to the reviewer for their feedback and very insightful questions. We address them below.
>
> ## Weakness 1&3 Paper writing
>
> The issues you pointed out are crucial for improving the paper's readability. We will carefully revise the paper. And I think it's necessary for me to rewrite the Introduction and Notation and Preliminaries sections. The specific revision outline can be referred to (in our global response part 1 'Paper Writing').
>
> For the specific problems you pointed out. Define $\Sigma^i$ as the strategy set of player $i$, where $\sigma^i\in\Sigma^i$. A strategy profile $\sigma=\mathop{\times}\limits_{i\in \mathcal{N}}\sigma^i$ is a collection of strategies for all players, $\sigma^{-i}=(\sigma^1,\dots,\sigma^{i-1},\sigma^{i+1},\dots)$ refers to all strategies in $\sigma$ except player $i$. $\Sigma=\mathop{\times}\limits_{i\in \mathcal{N}}\Sigma^i$ denotes the set of strategy profile, $\sigma\in\Sigma$. Define $u^i: \Sigma \rightarrow \mathbb{R}$ as the finite payoff function. We write  $u^i(\sigma^i, \sigma^{-i})$ for the expected payoff to Player $i$ if they select pure strategy $\sigma^i$ and all other players play the strategy profile $\sigma^{-i}$. Let $\sigma_t^i$ be the strategy used by player $i$ on round $t$. The average overall regret of player $i$'s action $a^i$ at time $T$ is $\bar{R}_{T}^{i}(a^i)=\frac{1}{T} \sum u^{i}\left(a^i, \sigma^{-i}_t\right)-u^{i}\left(\sigma_t\right)$.
>
> The definitions of these notions are all in Appendix A and will be incorporated into the main text in the revised version. I'm sorry for the bad reading experience.
>
> ## Weakness 2  Some plots in Fig. 2 are addressed while the other are not mentioned
>
> The issue you pointed out is indeed significant. We briefly introduce the functions of the unmentioned pictures in the section 5.2. The experimental result of PAM is to indicate that although the convergence speed of CFR+ algorithms was considered to exceed CFR, there are still some specific games (such as PAM) where the convergence speed of CFR+ is slower than the vallina CFR. However our proposed MCCFVFP and CFVFP are not affected. And 50 Card 3 Action 3 Len Kuhn shows that our algorithm has good performance in different games. These explanations will be added to the revised paper.
>
> ## Q1: The results for tangled games are not so strong as for clear games. At the same time tangled games are intuitively much more complex than clear games, which shows certain limitations of the proposed method.
>
> This question is very valuable. We don't think there are limitations to our algorithm in most cases. Firstly, the  training metric of the algorithm in Figure 1 uses the number of iterations. In the caption of this figure 1, we also clearly pointed out that the time complexity of one iteration of the FP algorithm is $\mathscr{O}(|\mathcal{A}|) $ and that of the RM algorithm is $\mathscr{O}(|\mathcal{A}|^2)$. If you take this into consideration, there is no significant difference in the convergence speed between FP and RM.
>
> Secondly, in lines 137-145 of the paper, we used many examples to illustrate that the proportion of dominated strategies in large scale games may be very high. So for actual games (such as Texas Hold'em), MCCFVFP will be a more suitable method. Moreover, in the latest experimental progress, we have developed a commercial multiplayer Texas Hold'em solver based on the MCCFVFP algorithm, and this solver is already online. We tested the effects of the MCCFVFP algorithm and the MCCFR algorithm in 2/3/6player no-limit Texas Hold'em in this solver. In these settings, the convergence speed of MCCFVFP leads MCCFR by 40%-50% (In global response part 3 "Experiments").
>
> Finally, lines 190-196 of the paper (and Appendix G.1) state that the implementation of MCCFVFP is particularly simple. It only requires 2/9 of the operation time of MCCFR.
>
> Based on the above factors, we don't think the use of MCCFVFP will be limited. Moreover, in our experiments, no case indicates that the convergence speed of MCCFR is faster than that of MCCFVFP.
>
> ## Q2.1 Random games with 21845 nodes do not seem to be challenging
>
> We don't think this will affect the effectiveness of our algorithm. This part of the content only shows that our theory is still valid in extensive-form games. The reason why no larger-scale experiments were conducted is that this experiment needs to generate the entire game tree before running and store the reward of all leaf nodes in the memory. It cannot calculate the reward of each leaf node in real-time searching according to the game process like other games, so using python to expand the scale is a little bit difficult. But we believe this scale is sufficient to explain our point of view.
>
> ## Q2.2 In section 5.3, the numbers of stored information sets of both methods are equal.
>
> The issue you pointed out is correct. This is attributed to my negligence in code development. Thank you very much for pointing out this problem. In the original version of the paper, we used the Openspiel framework developed by Deepmind for development. In this framework, the final strategy was not pruned. So 299376 is the number of information sets of the entire game, not the number of actually explored non-dominated nodes. We have updated all the content of this part. As mentioned earlier, we have developed a Texas Hold'em GTO solver based on MCCFVFP and conducted a comprehensive re-experiment. For details, you can refer to global response part 3 "Experiments".

---

> ### Comment · Reviewer_KPRH · 2024-08-09
>
> Thank you for your answers. I've read the other reviews and the rebuttal. I raised my score.

---

> > ### Author Response · Authors · 2024-08-09
> >
> > Thank you for your response! We appreciate your constructive and positive feedback regarding our paper.

---

### Official Review · Reviewer_H6L6 · 2024-07-16

**Soundness:** 3
**Presentation:** 1
**Contribution:** 3
**Rating:** 5
**Confidence:** 4

**Summary:**

The paper introduces a new algorithm called Monte Carlo Counterfactual Value-Based Fictitious Play (MCCFVFP) for solving extensive-form imperfect information games. This algorithm combines the counterfactual value calculations of Counterfactual Regret Minimization (CFR) with the best response strategy of Fictitious Play (FP). The authors demonstrate that MCCFVFP accelerates convergence to Nash Equilibrium (NE) significantly faster than existing Monte Carlo CFR (MCCFR) variants, especially in games with a high proportion of dominated strategies. They highlight its superior performance in large-scale settings, such as two-player limit short deck Texas Hold’em poker, where the blueprint strategy developed by MCCFVFP outperforms that developed by MCCFR.

**Strengths:**

The paper introduces an interesting integration of CFR's counterfactual value calculations with FP’s best response strategy, creating an approach to accelerating convergence in Monte Carlo settings. This combination provides a new perspective on solving extensive-form games. The theoretical proofs and experimental results are robust, showing evidence of MCCFVFP’s good performance in both convergence speed and practical application to large-scale games. The proposed algorithm has implications for the field of game theory and artificial intelligence, particularly in developing efficient strategies for large-scale, imperfect information games. This can have practical applications in various domains, including poker and other strategic games.

**Weaknesses:**

I believe the biggest drawback of this paper is its writing. Only readers with a deep understanding of CFR can comprehend the content of the article relatively smoothly. I think the author needs to include more background information in the main text, especially since it is currently only 8 pages long, and NeurIPS allows up to 9 pages.

The article contains numerous undefined symbols and typos, which greatly hinder the reader's understanding. It gives the impression that it was written in haste and requires thorough proofreading.

Line 58: ****

Line 72: u^i has never been defined.

Line 107: L has never been defined.

Line 157: R_i^i

Line 175: The same sentence appears twice.

Line 192: 6x−Ittakes2

Line 279: Table 1 liner

...

**Questions:**

1）In Figure 2, the author only compared CFR+. I believe DCFR and PCFR should also be compared.

2）How sensitive is MCCFVFP to the proportion of dominated strategies in a game? Is there a clear threshold where it starts to outperform MCCFR?

3）Have you investigated how MCCFVFP performs in multi-player games beyond two players?

**Limitations:**

Yes

---

> ### Author Rebuttal · Authors · 2024-08-06
>
> Thank you very much for your meticulous reading of the paper and for raising very valuable new questions.
>
> ## Paper Writing
>
> Regarding the typos in paper writing, we will thoroughly revise the paper.
> 1. **** At line 58, there is a hidden GitHub link. In the camera ready version, this link will be displayed.
> 2. The definitions of these notions are all in Appendix A and will be incorporated into the main text in the revised version. Here, I will briefly explain the meanings of these symbols. Define $\Sigma^i$ as the strategy set of player $i$, where $\sigma^i\in\Sigma^i$. A strategy profile $\sigma=\mathop{\times}\limits_{i\in \mathcal{N}}\sigma^i$ is a collection of strategies for all players, $\sigma^{-i}=(\sigma^1,\dots,\sigma^{i-1},\sigma^{i+1},\dots)$ refers to all strategies in $\sigma$ except player $i$. $\Sigma=\mathop{\times}\limits_{i\in \mathcal{N}}\Sigma^i$ denotes the set of strategy profile, $\sigma\in\Sigma$. Define $u^i: \Sigma \rightarrow \mathbb{R}$ as the finite payoff function.  $L=\max_{\sigma\in\Sigma,i\in\mathcal{N}}u^i(\sigma)-\min_{\sigma\in\Sigma,i\in\mathcal{N}}u^i(\sigma)$ represents the payoff interval of the game. Let $\sigma_t^i$ be the strategy used by player $i$ on round $t$. The average overall regret of player $i$'s action $a^i$ at time $T$ is:
> $\bar{R}_T^i(a^i)=\frac{1}{T}{\sum}u^{i}(a^i,\sigma^{-i}_t)-u^{i}(\sigma_t)$
>
> 3. The errors at line 175, 192, and 279 will be corrected in the camera ready version.
> Indeed, as you said, I mistook the limit on the number of pages of the paper and placed some definitions and notions in Appendix A, which made our work less reader-friendly. I will re-adjust the writing order in the revised version and provide more detailed explanations of the basic definitions, the purpose, and the significance of the paper. My outline of the revisions is in our global response part 1 "Regarding Writing".
>
> ## Question 1 "DCFR and PCFR should also be compared"
>
> This is a very valuable suggestion. We have made extensive modifications to the experimental part. DCFR, DMCCFR, and PCFR have all been added as baseline algorithms. In addition to adding the comparison baseline, we added three new games: vanilla Kuhn, vanilla Leduc, and  10-Card 1-Action 1-Len Leduc. In larger-scale games (15-card 1-action 1-len Leduc / 50-card 3-action 3-len Kuhn / PAM), these newly added baseline algorithms are still inferior to the MCCFVFP and CFVFP algorithms proposed by us. You can refer to our global response part 3 "Experiments" for details.
>
> ## Question 2 "Is there a clear threshold where it starts to outperform MCCFR?"
>
> This question is very interesting and meaningful. Define $\mathcal{A^i_\text{nd}}\subseteq \mathcal{A^i}$ to represent the set of non-dominated strategies in a normal-form game. We proved that the threshold between tangled game and clear game is $|\mathcal{A_\text{nd}}|\le \sqrt{|\mathcal{A}|}$ (In global response part 2 "The definition of clear games"). In our toy problem (a normal-form game with 100 actions), it is exactly $10\%$. So, thank you very much for your suggestion, which prevented us from missing a guiding conclusion for the paper. The proof will be added to the camera-ready version of the paper.
>
> However, we can only find the threshold of clear-game in normal-form games. Discovering a clear threshold in extensive-form games is extremely complex. We consider this to be a very valuable study and will attempt to solve this problem in future work.
>
> ## Question 3 "Have you investigated how MCCFVFP performs in multi-player games?"
>
> The direction you proposed is very insightful. I would like to briefly restate the theory of our paper: CFVFP is a Blackwell Approachability method, which means that in multiplayer games, CFVFP, like CFR, can converge to a coarse correlated equilibrium [1].
>
> In the latest research, we developed a commercial Texas Hold'em solver based on MCCFVFP (already online), completing the performance of MCCFVFP in multiplayer games. In 3-player and 6-player Texas Hold'em subgame, the performance of MCCFVFP exceeds the traditional MCCFR algorithm by approximately 40%. For more results, you can refer to our global response part 3 "Experiments" for details.
>
> [1] Zhang H, Lerer A, Brown N. Equilibrium finding in normal-form games via greedy regret minimization[C]//Proceedings of the AAAI Conference on Artificial Intelligence. 2022, 36(9): 9484-9492.

---

> > ### Author Response · Authors · 2024-08-12
> >
> > If you have any new questions or still have doubts about our current answers, please feel free to communicate with us at any time. We will try our best to answer any questions you may have.

---

### Author Rebuttal · Authors · 2024-08-06

# Global Response to All Reviewers

We thank all the reviewers for the detailed comments and constructive feedback. We found that many reviewers pointed out the same issues, including the writing of the paper, the related problems of clear-game, and the experimental part. Here, we will explain the problems in these three aspects uniformly.

## Paper Writing

We will proofread our paper in the revised version to ensure there are no typos. We will:
1. Rewrite the Introduction sections of the paper, ensuring that the background and purpose are clearer to make the advantages/disadvantages of the algorithm more obvious.
2. Update the Notation and Preliminaries sections of the paper. Move the definitions of Normal-form Game and RM from Appendix A to the main text to ensure the readability of the paper and enable people who are not familiar with CFR to understand the theoretical basis of the paper.

## The definition of clear games

In the original paper, we defined a game with a non-dominated strategy proportion of less than 10% a clear game. This definition is not scientific. We try to find a more appropriate explanation theoretically. Define $\mathcal{A^i_\text{nd}}\subseteq \mathcal{A^i}$ to represent the set of non-dominated strategies in the game. We have theoretically proved that, in normal-form games, the convergence speed of CFVFP is $\mathcal{O}\left( L|\mathcal{A_\text{nd}}|/\sqrt{T}\right)$, and the convergence speed of RM is $\mathcal{O}\left(L\sqrt{|\mathcal{A}|}/\sqrt{T} \right)$ , where $L=\max_{\sigma\in\Sigma,i\in\mathcal{N}}u^i(\sigma)-\min_{\sigma\in\Sigma,i\in\mathcal{N}}u^i(\sigma)$ represents the payoff interval of the game. Now we define a clear game as follows: When the number of non-dominated strategies of a game satisfies:
$$|\mathcal{A_\text{nd}}|\le \sqrt{|\mathcal{A}|}$$
The game is called a clear game. So in a matrix game with 100 actions, this proportion is exactly $10\%$, which is consistent with the conclusion of our toy experiment in paper. We will add this proof to the appendix and more experiments to verify the reliability of this proportion.

## Experiments

All reviewers pointed out that our experimental results were not sufficient (including game size setting / number of players / baseline selection). This is a very important suggestion, so we added more experimental results to the content of Section 5.2 and Section 5.3 of the paper.

### Section 5.2

In terms of the comparison baseline, DCFR, DMCCFR[1], and PCFR[2] were all added to the experiment. In addition, we added three new games: original kuhn, original leduc, and 10-leduc. In these experiments, we can prove the core viewpoint of our paper:
1. DCFR/PCFR/CFR+ may defeat MCCFR/MCCFVFP in small-scale problems such as  vanilla Kuhn. However, as the game scale expands. The convergence speed of sampling based algorithms (MCCFR/MCCFVFP) will gradually exceed that of full-traversal algorithms (DCFR/PCFR/CFR+).
2. In our experiments, the convergence speed of MCCFVFP is consistently faster than that of MCCFR. This implies that although our algorithm might theoretically be less effective than MCCFR in tangled games, considering the acceleration in the implementation of our algorithm and the fact that large-scale games tend to be clear games, our algorithm has always outperformed MCCFR in our experiments.
The new experimental results can be found in Figure 1 of the provided PDF. The code used in the paper will also be updated.

### Section 5.3

We have great news. We developed a commercial Texas Hold'em GTO solver based on the MCCFVFP algorithm, and this solver is already online. This solver can handle up to 6-player, 52-card no-limit Texas Hold'em. The ability to process games has been enhanced from approximately 300k information sets in the original paper to a maximum of about 170M information sets. In the latest experiments, we used this Texas Hold'em solver framework to test the performance of MCCFVFP in multiplayer large-scale games. In the 3-player and 6-player Texas Hold'em sub-games, the performance of MCCFVFP surpassed the traditional MCCFR algorithm, leading by approximately 40%.

Our experiments were conducted on the sub-game (river) of Texas Hold'em. All players call the big blind in preflop, then check until the river, with uniform card distribution and a 25BB stack depth.

1. In a two-player Texas Hold'em game, we can directly represent the convergence speed through exploitability. The results can be referred to in Table 2 of the PDF. The public cards in the experiment are 5d 2s 9d 2c 7c. After abstracting the hands, there are 69 strength ranks. This sub-game has approximately 89k information sets, and the unit of exploitability is BB/100.

2. In multiplayer situations, the exploitability cannot be directly calculated. Therefore, we use the method of mutual battles to measure the strength of different algorithms. In the competition 1, MCCFVFP AI is randomly set as player $i$, and all other players except player $i$ are set as MCCFR AI. Define $r_1$ as the rewards of the MCCFVFP AI player at competition 1. The second setting is exactly the opposite. Player $i$ is randomly set as MCCFR AI, and the remaining players are set as MCCFVFP AI. Define $r_2$ as the rewards of the MCCFR AI player at competition 2. By comparing $r_1$ and $r_2$, we can roughly compare the convergence speeds of different algorithms. The results can be referred to in Table 3 of the PDF.

Full details will also be included in the camera-ready version.

[1] Brown N, Sandholm T. Solving imperfect-information games via discounted regret minimization[C]//Proceedings of the AAAI Conference on Artificial Intelligence. 2019, 33(01): 1829-1836.

[2] Farina G, Kroer C, Sandholm T. Faster game solving via predictive blackwell approachability: Connecting regret matching and mirror descent[C]//Proceedings of the AAAI Conference on Artificial Intelligence. 2021, 35(6): 5363-5371.

---

> ### Author Response · Authors · 2024-08-11
>
> In Table 2 of the PDF, the JsQsJs3h2h in the third row should be KsQsJs3h2h. This is hereby corrected.

---

### Decision · Program_Chairs · 2024-09-25

**Decision:**

Accept (poster)

**Comment:**

The paper introduces a new algorithm for solving extensive-form imperfect information games. The algorithm combines, in a novel way, Monte Carlo counterfactual regret minimization and fictitious play, achieving faster convergence to Nash Equilibrium in the experiments. The experimental results are justified by theoretical analysis of the algorithm. All reviewers agreed that the proposed algorithm is novel and interesting, and the results are promissing.

That said, most reviewers expressed concerns about writing and presentation, providing recommendations and guidelines about improvements. Also, some reviewers were concerned that comprehending the paper requires a non-trivial background in the literature on game theory and state of the art algorithms in the field.

The authors provided detailed feedback on the reviewers' concerns, and suggested constructive ways for preparing the paper for publications, and the reviewers appeared to be convinced by the authors' feedback. Based on the expressed concerns and proposed improvement and corrections, I recommend acceptance as a poster.